# Unlocking the Power of Mentoring: A Comprehensive Guide to Evaluating the Impact of STEM Mentorship Programs for Women

Elke Wolf [1],* and Stefanie Brenning [2]

1    Department of Engineering and Management, Hochschule München University of Applied Sciences, 80335 Munich, Germany
2    Department of Tourism, Hochschule München University of Applied Sciences, 80335 Munich, Germany
*    Correspondence: elke.wolf@hm.edu

**Abstract:** Although mentoring programs for female STEM students are often carried out with a great deal of passion on the part of program managers and mentors, robust results on their effects are often missing. However, regular evaluations are indispensable for an efficient allocation of resources towards gender balances in STEM. To accomplish this requirement, empirically valid and easy-to-use evaluation concepts are needed. We therefore develop an evaluation concept which corresponds to a Logic Chart, capturing three levels of expected effects (output—outcome—impact). On each level of impact, we derive a set of success indicators that can be measured by qualitative methods. A major advantage of our evaluation design is that the effect of a mentoring program can be observed directly after the end of the program. Furthermore, the results provide information about different channels of impact (e.g., reduced stereotypes or increased self-efficacy) and hence offer concrete indications for the further development of the program.

**Keywords:** evaluation; mentoring; STEM; female students

## 1. Introduction

Female students in STEM courses, especially in the fields of information technologies and engineering, face particular challenges. They usually know few female role models and are confronted with prejudices and stereotypes about their technical competence, as well as a professional habitus with male connotations (e.g., Minks 2004; Paulitz 2014). These manifest as assumptions about which gender is particularly successful in STEM (Ihsen and Ducki 2012), who has specific technical or social competencies (e.g., Gilbert 2008), who brings special interest to the field of study (e.g., Ihsen 2006), or who will enter an adequate profession and be successful there (Ihsen 2010; Derboven and Winker 2010). As a result, the performance of women in STEM might be impaired due to the stereotype threat (Shapiro and Williams 2012). All these stereotypes come at a high cost, not only to individual women, in terms of lost income and missing career options, but also to the economy, in terms of skills shortage and lack of innovation (World Economic Forum 2019).

Mentoring programs have the potential to empower female students, reduce stereotypes and the stereotype threat, and introduce an array of attractive career opportunities for women in STEM. In fact, there is evidence that mentoring programs for STEM students can help increase students' self-efficacy, provide personal networks, and reduce the risk of stereotype threat and dropout among first-year students and especially among women (see, among others, Hernandez's (2018) survey for the United States). However, evaluations of mentoring programs for female STEM students are very rare (see, among others, Hernandez et al. 2017; Dennehy and Dasgupta 2017; Byars-Winston et al. 2015; Reid et al. 2016; MacPhee et al. 2013; Stout et al. 2011). Despite the abundance of mentoring programs at German universities, this is also true for Germany (see, among others, Callies and Breuer 2010; Hessisches Koordinationsbüro des MentorinnenNetzwerks 2010; Höppel 2016; Stöger

et al. 2013). Moreover, these studies and evaluations vary considerably in their design and in the quality of their designs. Therefore, it is often difficult to assess whether the outcomes reported are supported with sufficient evidence. This is particularly a dilemma for studies of mentoring programs, as participation is voluntary and thus less formal than mandatory modules in the curriculum, and there is also considerable variation in the number of contact hours between programs and even between participants in the same program. Thus, for many programs, the random assignment of participants to treatment and control groups, often considered the gold standard in research design, is logistically infeasible for many programs, could alienate learners, and could compromise the validity of certain studies. Wolf and Brenning (2019) reveal, in their meta-evaluation of STEM projects for female students in Germany, that only singular studies use a control group or a pre-post comparison to quantify the causal effect of a program (these include Findeisen 2006; Leicht-Scholten and Wolf 2009; Stöger et al. 2013, 2016, 2017). Thus, there is not only a lack of quantity, but maybe also a lack of quality in evaluations—a finding that ultimately applies to almost all equality-promoting measures at universities and beyond (Kalpazidou Schmidt et al. 2017; Löther 2019).

Unsurprisingly, little research has been conducted on practical concepts for evaluating these educational offerings in Germany. This becomes apparent by the fact that the anthology on the 10th anniversary of the Federal Association for Mentoring in Science (Forum Mentoring e.V.) discusses many practical reports and contributions on quality assurance, but does not include any contributions to the evaluation of mentoring programs (Petersen et al. 2017). Looking at specific programs for female students in STEM, the evidence becomes even thinner. Since evaluation methods from the textbook cannot be transferred so easily to the evaluation of mentoring programs, there is a lack of application-oriented concepts to measure the impact of these gender equality-promoting measures. In the USA, the National Academy of Science, Engineering and Medicine therefore convened a committee of experts in 2017 to develop guidelines for evaluating mentoring programs for STEM students (National Academy of Science, Engineering and Medicine 2019). Although there are some handouts on how to prepare evaluations of STEM mentoring programs (National Center for Women & Information Technology 2011; Nickolaus and Mokhonko 2016; Nationales MINT Forum 2018; Kingsley 2020), there is still an unmet need for a valid and simple evaluation concept in practice. While all guides provide helpful insights on how to measure the effects of mentoring programs for female STEM students, our approach stands out for its evidence-based foundation of the impact mechanism and the evaluation methods.

To close this gap, we have developed an evaluation concept that is flexible enough to apply to almost all mentoring programs for female STEM students, aiming at successfully guiding female STEM students through their studies and into their careers. It is based on the above-mentioned empirical findings that women in STEM fields must overcome numerous hurdles—some of them "invisible"—in order to be successful in their studies and careers. Mentoring programs are designed to remove these hurdles or rather, to empower the participants to overcome these obstacles. Thus, a successful mentoring program brings down at least one of these hurdles. Our evaluation concept measures the contribution that participation in a mentoring program makes to overcoming these hurdles using quantitative methods. To derive the indicators of success, we use a logical impact model that describes both the long-term goals and the necessary intermediate steps to get there. The relationships between the various stages of the impact model are scientifically founded and based on theoretical approaches and empirical findings to explain change (change theory). This theory-based evaluation design promotes understanding not only whether a program works or not, but also how it works, that is, which of the various obstacles broke down. Hence, continuous monitoring of STEM projects with the help of this approach also offers concrete starting points for the further development of the mentoring programs, and thus contributes to quality assurance (Reinholz and Andrews 2020). The development of the evaluation concept was accompanied by three empirical pilot studies

in which the concept was applied to very different mentoring programs at universities of applied sciences and was continuously improved.

In the following, we present the basic idea of our evaluation concept (Section 2). In Section 3, we expound the underlying change theory of mentoring programs for STEM students. We derive various indicators at the level of output, outcome, and impact, and provide concrete survey questions for the empirical implementation. We then discuss different ways to identify the causal relationship between participation in a mentoring program and the intended goals (Section 4). Finally, we give guidance on how to interpret the empirical results (Section 5) and conclude with a summary and an outlook.

## 2. The Basic Idea of Our Evaluating Concept for Mentoring Programs

Since mentoring programs for female STEM students typically intend to contribute to a successful completion of studies and facilitate entry into professional life, the effectiveness of this measure typically only becomes apparent some time after participation in the program. Therefore, evaluating a mentoring program after graduation or even later would seem to be most straightforward. Whether the mentoring program prevents students from dropping out and actually improves the career opportunities of women in STEM professions could thus be determined with the help of a long-term panel analysis about the participants' further academic and professional career.

However, this approach has four serious disadvantages. First, gathering longitudinal data is arduous and costly due to the difficulty of tracking former mentees after several years. Second, the response rate to surveys after graduation will be very low. The longer the time since the events took place, the lower the commitment to contribute to the evaluation of the mentoring program. Third, it must be recognized that academic success or a successful career entry in the STEM field depends on various factors. Theoretically, the influence of determinants outside the mentoring program could be accommodated with the help of multivariate analyses, but, in practice, this option is hardly feasible due to typically small numbers of participants and the long list of potentially relevant control variables. Identifying the effects of a single key event, such as a mentoring program, would not be possible that way. Finally, no insights into the mechanisms of mentoring programs could be gained by focusing on the long-term impact.

Our evaluation concept is, therefore, based on an assessment immediately after participation in the program. At this point, it is already possible to observe how the mentorship influences the determinants of women's academic and professional careers in STEM in the short term. But what are the relevant determinants of success? Or, asked the other way round: what are the challenges and obstacles that prevent the success of female STEM students? Fortunately, this question has already been researched extensively (Beck et al. 2021; Blackburn 2017; Steffens and Ebert 2016). One of the biggest barriers to STEM is its continuous perception as being male-dominated (Lee 2008) and, as a result, there is a low sense of belonging and a working climate that women perceive as chilly (Hughes 2014; Miner et al. 2019). Furthermore, women receive less encouragement to develop their math and science skills from their family, friends, or teachers. The underlying stereotype that women are endowed with poorer mathematical skills is a myth that persists obstinately and often impairs the actual performance of girls and women (Rea 2015; Shaffer et al. 2013; Shapiro and Williams 2012). Finally, women in STEM have less opportunities to get inspired by same-sex role models and benefit from their experiences, e.g., insights with regard to the compatibility of family and career. Taken together, these realities mean that women need an extra dose of perseverance, resilience, and enthusiasm to overcome these obstacles and successfully complete a STEM degree.

Of course, some of the obstacles, such as the attitude of the family environment, the peer group, or the teachers, cannot be changed by a mentoring program. However, good programs have the potential to mitigate at least some of the hurdles female students have to deal with. Examples of this are stereotypical thinking patterns that can be dissolved or softened through contact with role models (Shin et al. 2016; González-Pérez et al. 2020). In any

case, a mentoring program can build resilience and compensate for the lack of personal relationships. We therefore develop a comprehensive and theoretically sound impact model to derive the determinants of academic and career success of female STEM-students that could be changed by a mentoring program (Section 3). This approach provides a list of success indicators on different stages of impact that can be measured by quantitative methods.

Given the emerging criticism of the traditional inference of causality for reasons of practicality, high cost, and ethical hurdles, other ways of measuring impact are increasingly discussed in evaluation research (Cook 2007; Scriven 2008; Gates and Dyson 2017). Our evaluation approach takes up these reflections and uses a probabilistic, rather than deterministic, approach to describe impact. Our approach considers different impact pathways as contributions to the overall outcome (Cartwright 2007; Steiner et al. 2009). In this sense, there is not only one causality, but a multitude of causal relationships that are underpinned by a theoretically sound impact model. Furthermore, looking at the different partial effects counters Müller and Albrecht's (2016) criticism that evaluations focusing on long-term impact only do not generate insights into the mode of action of the measures studied. Our approach has the advantage that it provides program managers with information on how the measures are affecting the various obstacles, which is invaluable for the further development of the offers. Due to the general difficulties in attributing changes, with regard to gender equality, to individual programs and not to larger societal trends and influencing factors (Kalpazidou Schmidt et al. 2017; Löther 2019; Palmén and Kalpazidou Schmidt 2019), this approach represents an insightful alternative to the one-dimensional impact analysis.

In order to make the practical application of our evaluation concept as simple as possible, we also present selected questions to capture these determinants. Additional questions are published in Wolf and Brenning (2021a, 2021b). Note, however, that every questionnaire must be adapted to the specific context of the mentoring program and validated instruments should be applied to measure the relevant indicators.

## 3. The Change Theory of Mentoring Programs of Female STEM Students

Mentoring programs to support female students in STEM give mentees the opportunity to receive support in their studies and career planning from a mentor in their field. Typically, mentoring programs do not include subject-specific classes and thus do not serve to close knowledge gaps. Seminars and workshops within mentoring programs are, rather, intended to deepen the soft skills and key qualifications of female students. Regular networking meetings and informal one-to-one meetings between mentee and mentor help to share the experience of the mentors. The students not only gain insight into the career steps of their mentor, but also get to know many role models. As much as the many mentoring programs for female STEM students differ, they all pursue the goal of supporting female students in their studies, reducing the risk of dropping out, encouraging them to choose a career within the STEM field, and supporting the occupational advancement of women in STEM fields (Beck et al. 2021; Wolf and Brenning 2019).[1] But what are the pillars of success for mentoring programs? What mechanisms of action need to be triggered to achieve these goals?

In the following, we describe the ideal-typical effects of a standard mentoring program based on a logical impact model. The so-called Logic Chart visualizes the relationships between activities, program content, and short-, medium-, and long-term impacts, and helps to effectively plan, implement, and evaluate interventions (W.K. Kellogg Foundation 2004, III). According to this, the events and activities within the mentoring program generate immediately measurable results (output), short-term effects (outcome), and long-term effects (impact) (Döring and Bortz 2016, p. 984). Of course, several partial effects are relevant at each stage of impact. Since the needs and personal challenges of the participants are diverse, the success of a mentoring program is made up of many small puzzle pieces. Each of these partial effects can contribute to the final goal of the mentoring program and can thus be used as an indicator for the program's success. Our evaluation concept

captures a large set of partial effects at all three stages of this Logic Chart and hence allows a comprehensive assessment of a program. Consequently, our evaluation approach considers that the causal effect of a mentoring program is neither linear nor monocausal. Rather, many different changes are needed to improve the decisions and successes of women in STEM professions (Löther et al. 2021).

Figure 1 describes the modes of action of mentoring programs for female STEM students. Our impact analysis relies on the Logic Chart developed by Löther and Girlich (2011), which adopts the general logical model to analyze the effects of different measures for study and career orientation for female students. We apply this Logic Model to mentoring programs for female STEM students.

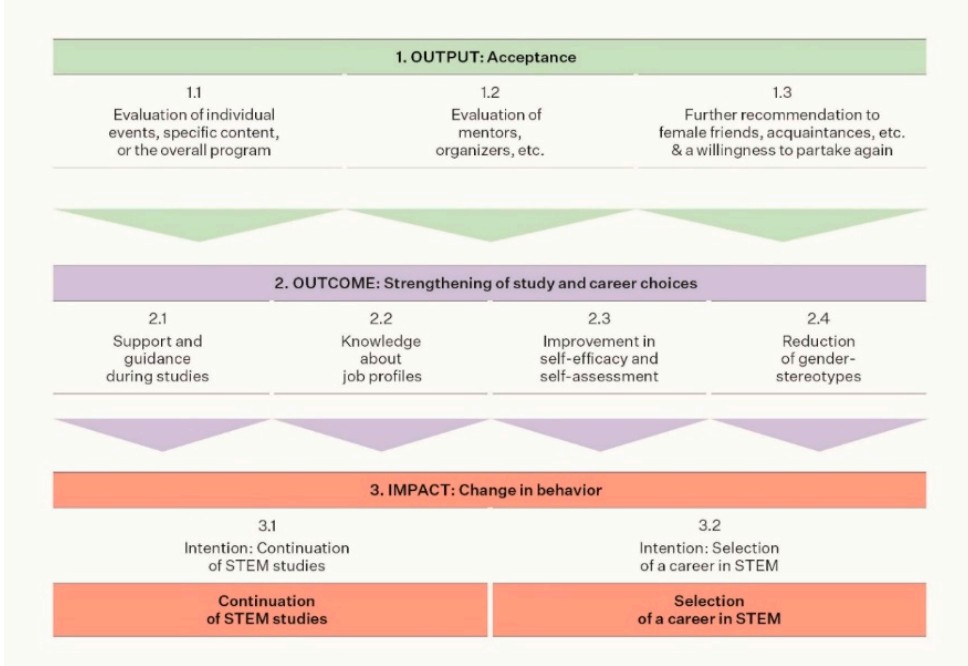

**Figure 1.** Logic Chart illustrating the intended effects of mentoring programs. Note: Own illustration.

### 3.1. The Output of a Mentoring Program

The first stage (1. Output) captures the acceptance of the mentoring program by female STEM students. A positive assessment of the actions in place is the basic prerequisite for further effects on the next level of impact. This is shown by the participation and assessment of the events (e.g., kick-off event or workshops) (1.1), the service orientation of the organizational processes, as well as the quality of the mentoring relationship and the mentor (1.2). Furthermore, acceptance includes the extent to which the participants would recommend the mentoring program to others and/or participate in the program again (1.3).

Helpful questions to measure the acceptance of the mentoring program are:

- Which events of the mentoring program have you attended at least once? (1.1)
- How satisfied are you, with regard to the following organizational aspects? (Online Information, registration process, communication with the program manager, organization of the mentoring program) (1.1)
- How often have you had personal meetings with your mentor? (1.2)
- How satisfied are you with the following aspects of the mentoring relationship?
- (personality and sex of the mentor, professional fit with the mentor, spatial distance to the mentor, commitment of the mentor, discussed topics) (1.2)
- Will you recommend the mentoring program to fellow students? (1.3)
- Can you imagine participating as a mentor in the future? (1.3)

### *3.2. The Outcome of a Mentoring Program*

The second stage (2. Outcome) describes the inclination to choose a career path in STEM. In theory, all activities within the mentoring program strengthen the study and career choices of the students. This is the case if mentors provide students with concrete support and guidance in the course of studies (2.1). They could, for example, assist in the initiation of a practical semester or the final thesis. Numerous studies show that a lack of support and orientation during studies, a lack of social integration, and a lack of practical relevance are factors that encourage students to drop out (Meyer et al. 1999; Meinefeld 1999; Pohlenz and Tinsner 2004). Bilateral exchange with a mentor or discussions in seminars can help to mitigate adverse study conditions for women in STEM, thus contributing to higher study satisfaction and reducing the likelihood of dropping out (Fleischer et al. 2019; Fischer et al. 2020).

Furthermore, knowledge gaps about possible job profiles in STEM are also relevant reasons for dropping out (Heublein et al. 2010). Mentors can fill this gap by sharing information about different job profiles and employers in the STEM field or by arranging an internship in a specific company or sector. Ultimately, all contacts and information gained through the program help young women to get a big picture about their options and to develop their own career (2.2). According to the expectancy-value theory by Eccles and her colleagues, expectations of success and subjective task values are presumed to shape achievement-related choices and persistence, such as career choice (Eccles 2005). Whether a job task is considered valuable depends, among other things, on the utility, relevance, and meaningfulness attributed to it. Mentors can make an important contribution to the appreciation for tasks in STEM if they place the achievements of STEM professions in a wider context of societal challenges, such as climate change, mobility, or demographic change, and emphasize the team-oriented working conditions. This perspective refreshes the image of STEM professions and matches women's preference for jobs where they work with people (Su and Rounds 2015) and for careers that pursue community goals or helping others (Diekman et al. 2010; Edzie 2014). Hence, a comprehensive knowledge about job profiles can help women to opt for STEM (2.2).

The academic success of female students is also largely determined by their self-efficacy and self-assessment (2.3). The results of Weinhardt (2017) show that female students assess themselves worse compared to male students in mathematics, even with the same grades. Furthermore, Correll (2001, 2004) shows that girls are more likely to attribute their success in mathematics and science to luck and effort, while boys view success as a consequence of their competence. Especially for STEM students, performance problems represent one of the most common reasons for dropping out (Heublein et al. 2010; Seemann and Gausch 2012). Therefore, we conclude that female STEM students are more likely to drop out of university if they have a lower self-assessment of their mathematical and technical skills. In contrast to self-assessment of one's own abilities, which result from personal experiences, self-efficacy includes expectations that relate to the successful accomplishment of future challenges (Bong and Skaalvik 2003). According to Bandura (1997), self-efficacy describes a person's belief in his or her ability to perform certain tasks successfully. If, for example, this basic belief in one's ability to cope with the demands of a STEM study or profession is less pronounced for female students because they are less confident in their competencies (Correll 2001, 2004; Fellenberg and Hannover 2006) or expect particular difficulties as women in a male-dominated environment, this could call into question women's academic success and their choice of a STEM profession. Participation in empowerment workshops can break these inhibiting patterns of thinking and increase the mentees' self-efficacy and self-assessment. In addition, the mentors' experiences can help them gain a better understanding of gender-specific career obstacles and helpful information about the rules of the game as they are lived out in companies. This knowledge also increases self-efficacy and reduces the risk of dropping out of university (Fellenberg and Hannover 2006).

Of course, gender stereotypes also influence the continuation of STEM studies and the choice of a job in the STEM field in different ways (2.4): On the one hand, negative

stereotypes regarding women's mathematical and analytical abilities can reduce the corresponding performance of female students, as described in the literature on stereotype threat (see among others Spencer et al. 1999; Pennington et al. 2016). In turn, women's interest in STEM may decline (Shapiro and Williams 2012; Master 2021), leading to poorer assessments of their own abilities along with the above-mentioned consequences. Second, many STEM disciplines are associated with masculinity, and competencies in STEM are perceived as unattractive and unfeminine (Miller et al. 2018; Thébaud and Charles 2018; Cheryan et al. 2015; Ertl et al. 2014; Kessels and Hannover 2006; Glick and Fiske 1999). As a result, women feel less of a sense of belonging in STEM and are less confident that their effort will bear fruit (Strayhorn 2012). Following Festinger's (1957) theory of cognitive dissonance, everyone strives for consistency in his or her own opinions and attitudes and tries to reduce dissonance between incompatible cognitions. If there are discrepancies between one's own gender-specific self-concept and the stereotypes about women in STEM, women tend to distance themselves from this professional image and tend to choose "girl-typical" fields of study and occupations that are not in the STEM field (Kessels 2015). Doing so, they reduce the cognitive dissonance experienced as uncomfortable. Again, personal contact with role models from the STEM field can mitigate the stereotype threat, increasing students' self-efficacy and making them more likely to consider a career in a male-dominated environment (Kosuch 2006; Lins et al. 2008; Solga and Pfahl 2009; Shapiro et al. 2013; Stout et al. 2011).

Measuring the level of inclination to choose a STEM field is not straightforward because most people are not aware of their self-efficacy or stereotypes.[2] Options to measure the four partial effects are:

- Participation in the mentoring program helped me to get . . . (e.g., motivation for my studies, support for the practical semester/final thesis, access to networks, to know my strengths and weaknesses) (based on Höppel 2016). (2.1 and 2.3)
- How satisfied are you, overall, with your studies? (2.1)
- Are you well-informed about possible jobs you can take up after graduation? (2.2)
- Participation in the mentoring program helped me to get . . . (e.g., to know role models, insights into the everyday working life of a professional, access to networks, a better idea of the kind of company I want to work in later, a better idea of which professional field I want to work in later, an understanding of the obstacles women face in their careers, information about the "rules of the game" in companies) (based on Höppel 2016). (2.2)
- If I take up an activity in my professional field, then . . . (e.g., I can contribute to solving important social problems, I need good social competence, I can combine my job with my own family, I work a lot in a team, . . .). (2.2)
- How confident are you that you can cope with the demands of your studies? (Fellenberg and Hannover 2006). (2.3)
- After completing my studies, I can imagine . . . (e.g., leading a project team, mastering negotiating (with men) confidently, working in a male-dominated environment, to have a female supervisor). (2.3 and 2.4)
- Do you agree with the following statements? (2.4)
- Most women know well about '. . .' (put in a specific STEM field).
- Most men know well about '. . .' (put in a specific STEM field).
- I can identify well with my field of study. (2.1 and 2.4)

### 3.3. The Impact of a Mentoring Program

On the third level of our Logic Chart (Figure 1), the impact measures the long-term effects (Döring and Bortz 2016, p. 985) that are generated by changed behavior or a change in the private or professional situation of the participants. This includes the completion of STEM studies (3.1), as well as the successful entry into a STEM profession (3.2). Since these effects often reveal themselves years later, the impact could alternatively be measured by the expressed intention.

Ajzen (1991) argues that behavioral intention is determined by attitudes, which are influenced by experience and external factors. That is, behavioral intentions are indirectly modifiable by exogenous events and are directly related to actual behavior. Kim and Hunter (1993) show in their meta-analysis that attitudes explain 50 per cent of the variance in specific behavioral intentions and these, in turn, explain 30 per cent of the variance in actual behavior. Armitage and Conner (2001), in their meta-evaluation of 185 different studies, also conclude that intentions and self-reported predictions are good predictors of behavior. Even though the theory of planned behavior aims to explain the short-term relationship between intention and action, Randall and Wolff (1994) show that the strong correlation coefficient is maintained even up to a temporal distance of 15 years. Stöger et al. (2013) therefore use study choice intentions as one of the indicators of success for the effectiveness of the program in their evaluation of a one-year online mentoring program for eleven- to 18-year-old schoolgirls. Lent et al. (1994) also argue that the theory of planned behavior can be applied to study and career decisions, implying that this decision is also influenced by the prior behavioral intention. A change in study and career choice intentions in the desired direction thus increases the probability of a certain career choice and can consequently be interpreted as a success of the STEM program for women.

While measuring the actual impact requires tracking the mentees for years, the intention to complete the degree in a STEM subject or to choose a job in the STEM field can be captured right after the end of the mentoring program with the following questions[3]:

- I am seriously thinking of dropping out of university/my doctorate (Fellenberg and Hannover 2006). (3.1)
- I am seriously thinking of taking up a master's program after completing my bachelor's degree. (3.1)
- After completing my studies, I will take up a profession in the STEM field. (3.2)
- In the course of my career, I will take a leadership position. (3.2)

## 4. The Empirical Evaluation Design

The goal of a valid evaluation is to establish a causal relationship between participation in a mentoring program and the intended goals and intermediate goals of the program. Based on the Logic Chart presented above, the treatment effects $\delta^j$ of a mentoring program (j represents the various partial effects illustrated in Figure 1) are measured using the differences of the various indicators on the different stages of the Logic Chart $Y_{i1}^j$ in case of participation (T = 1) and non-participation (T = 0). $X_{i1}$ describes further individual factors of influence at time t = 1 (e.g., school grades or gender-specific socialisation) and $Z_1$ general determinants (e.g., the wage level or the prestige of an occupation) on personal study and career choices:

$$\delta^j = Y_{it}^j(X_{i1}, Z_1, T = 1) - Y_{it}^j(X_{i1}, Z_1, T = 0),$$

The challenge of evaluation is that a person *i* can never be observed in both states at the same time (Wooldridge 2013). To overcome this problem, different evaluation designs can be used, which differ by identifying causality, and each have advantages and disadvantages in planning, implementation, and validity. In the following, we present the possibilities and restrictions of different evaluation designs. We begin with the evaluation approach, which—under optimal conditions—is best suited to identify the causal effects of STEM projects and then discuss the more pragmatic approaches.

The experimental study design is intuitive but demanding in practice. It is based on the idea that the non-observable counterfactual state $Y_{it}^j(X_{i1}, Z_1, T = 0)$ is measured with the help of a control person $Y_{it}^j(X_{j1}, Z_1, T = 0)$ or a control group. However, this method only produces valid results if the treated persons and the members of the control group do not differ regarding the relevant person-specific characteristics $(X_{i1} = X_{j1})$. In order to ensure that the two groups are ultimately comparable, female aspirants for the mentoring program would have to be admitted by random selection, which is typically not feasible due to ethical concerns or the limited number of interested women. If we

were to use only female students who do not participate in the STEM mentoring program as a control group, this requirement would hardly be met. According to Brenning and Wolf (2022), the participants of career-promoting measures have a higher motivation and career inclination than female students who do not participate in a measure. As a result, a simple comparison of participants with non-participants would overestimate the effects of said measure. Theoretically, multiple interviews with comparison and control groups (difference-in-difference approach) are a way to eliminate differences in observable and unobservable determinants and to measure the estimated effects precisely and without bias (Wooldridge 2013). In practice, however, this procedure is very laborious. Moreover, the results can still be biased due to small sample sizes, the Hawthorne effect, or the impossibility of "hidden" treatments, as well as placebo effects (Colbjørnsen 2003; Deaton and Cartwright 2018).

The quasi-experimental design, which follows the logic of an experiment but does not randomize the selection of treatment and comparison groups, is the study design with the next-best informative value, with regard to measuring causal effects. Although the effects of a lack of randomization can be mitigated in large samples with the help of multivariate statistical procedures and/or the matching of observations of the experimental and control groups, there is still the risk of distorting the effects of the treatment through unrecorded third-party variables (Diekmann 2012). Furthermore, mentoring programs typically take place in small groups, so that large samples are rather rare. Alternatively, in the case of small samples, the sequential treatment could be applied, in which participants of other/later measures (e.g., students from the waiting list of the measure) serve as a control group (Duflo et al. 2007). However, this presupposes that there is a waiting list, or that the participants in the mentoring programs are known in advance for a relatively long time.

Pre-experimental designs forgo the comparison with a control group and survey the mentees either before and after participation in the program, or only after the program (Diekmann 2012). However, since a one-time survey cannot measure changes in the determinants of academic and career success, causal effects cannot be demonstrated this way. In contrast, a before-after comparison without a control group can only quantify the causal effect of a measure on the determinants of study, and career choice, if the outcome after non-participation $Y_{i1}(X_{i1}, Z_1, T = 0)$ is identical to the outcome before non-participation $Y_{i0}(X_{i0}, Z_0, T = 0)$. Theoretically, however, it cannot be ruled out that other impulses for the study program, and in career planning, arise during the observation period (e.g., through a new lecturer, a new module, or a project within the framework of the regular study program). In this case, the observed changes would not have been triggered by the mentoring project, but by changes in the personal influencing factors $X_{i1}$ or general trends (changes in the determinants $Z_i$). Kalpazidou Schmidt et al. (2017) also point out that, in the field of gender equality, it is fundamentally difficult to assign changes to a specific program and thus to rule out the possibility that general context-related trends change reality. However, the shorter the observation period, and the more stable the other determinants $Z_1$ of study and career choices are, the less significant this problem becomes. Furthermore, the success indicators can also be affected by short-term fluctuations throughout the semester. It is conceivable, for example, that study satisfaction and self-assessment are higher at the beginning of the semester than just before or during the exam period, when the stresses of studying are significantly higher. These effects can distort the results of a simple before-and-after comparison, although the problem could be mitigated by the clever selection of survey timing.

In view of the temporal persistence of gender stereotypes in the context of study and career choice, it is not to be expected that (within the typical project duration of a mentoring program) serious distortions will arise due to rapid social change. Furthermore, it can be assumed that the personal influencing factors $X_{i1}$ (e.g., the personal competencies, the gender-specific socialization of the female students, and associated influences, such as the attitudes of friends and family) are mostly very stable over time, so that no serious changes

are to be expected during the relatively short observation period, and a simple before-and-after comparison can certainly lead to comparatively reliable results. To conclude, this study design can be implemented in practice without much effort and therefore represents a good and pragmatic solution.

Nonetheless, the survey results must be interpreted with caution. For example, the survey of female students, which often takes place directly after participation in the measure, increases the probability of socially desirable answers. This is especially true if the participants fill out the questionnaires in the presence of the project coordinators, whose project they may not want to rate badly. Even if the students do not intend it, the Hawthorne effect, which is caused by the special attention given to the participants, leads to an overestimation of the measured effects in a pre-post survey (Colbjørnsen 2003). However, this bias is inherent in all surveys and can only be mitigated by designing a professional interview situation, ideally without interviewers who are somehow personally involved in the mentoring program. On the other hand, it is conceivable that the effect of the support measure is greater directly after the last event than at a later point in time, so that the long-term effects in particular may be overestimated.

In summary, in our view, before-and-after surveys provide quite valid information for recording the effects of mentoring programs on the study and career choices of female STEM students, even without comparison with a control group. Nevertheless, when designing a program or an evaluation, project managers and evaluators should explore all possibilities to collect complementary information on non-participants. Even if random assignment is not fully possible, these data can provide valuable information on the selectivity of participants, which, in turn, might have implications for addressing the intended target group in the marketing. In practice, however, theoretically more valid evaluation designs may not be applicable due to the small number of cases and the project context.

## 5. Collecting and Interpreting the Data

For a comprehensive assessment of an intervention, evaluations should capture impacts at all three levels of our Logic Chart. Questions on the content and organization of the mentoring program (output) can, of course, only be asked in the final questionnaire and primarily serve to fine-tune the content of the program and ensure acceptance, as well as the subjective satisfaction of the participants. Questions on the success indicators at the outcome and impact level must be asked in both questionnaires, as conclusions about the effects of the program can only be drawn by comparing the answers. Typically, responses to questions about the various outcome and impact indicators are captured using Likert scales, which offer respondents the opportunity to express their exact degree of agreement with a statement (see questions in Section 3.2). Since attitudes or behavioral and personality traits (e.g., gender stereotypes or self-efficacy) often cannot be addressed by only one statement, multiple items are used in the questionnaire and then aggregated into an appropriate indicator. To compare before and after answers (e.g., of the Likert scale), evaluators can either decide between or combine three descriptive measures: (1) comparison of group mean values, (2) differences in the individual answers (illustrated by a frequency count of positive and negative changes)[4], and (3) difference in the share of agreement (defined as the share of individuals who (totally) agree with the statement).

Whatever metrics are calculated, the question of how to interpret the data is still open and not straightforward. What does the data tell us about the success of a mentoring program? Is the mentoring program fruitful, even if only some of the success indicators change in the desired direction? Do all indicators have the same importance for the assessment of the mentoring program? There are no clear answers to these questions. Basically, it is much easier and more harmless to focus on the partial effects, instead of trying to make a general and comprehensive assessment of the whole program. If at all, the weighting of the partial results should always be in line with the goals of the respective mentoring program, as well as the goals expressed by the mentees. In the end, the evaluator

must decide (in alignment with the goals) which criteria are relevant for the success of the mentoring program.

However, undetectable effects on one or more dimensions do not necessarily mean that the project is totally ineffective. Even if only one indicator at the outcome level is switched in a positive way, long-term changes in behavior are nonetheless conceivable. For example, the mentees' assessment of the compatibility of work and family life in a STEM job (see 2.2. in the Logic Chart) may be revised thanks to a mentor with family duties. This insight could make the decisive contribution to the choice of a STEM profession. Concerning stereotypes (see 2.4. in the Logic Chart), activities to increase self-efficacy, whether through personal exchange with the mentor or personality-building workshops with other mentees, curb the power of stereotype threat and may turn the decision towards STEM.

What does the deterioration of an indicator say about the success of a mentoring program? First, it should be kept in mind that, on average, more self-confident and career-oriented women opt for the mentoring program. That is, most mentees already start the program with better grades, higher scores in terms of career orientation, and greater self-efficacy and interest (Brenning and Wolf 2022; Stöger et al. 2016). Due to this positive selection, the fruits of success tend to hang high. Secondly, negative effects may also be triggered by the fact that the concrete examination of the situation of women in STEM professions makes the potential hurdles in the career process even more visible, so that the attitudes of the female students might become more pessimistic, albeit more realistic. Nevertheless, undesirable changes should be taken as an opportunity to take a closer look at the suitability of the mentors (in terms of gender competence and knowledge of how to promote mentees) as well as the quality of the mentoring relationship, as these factors also contribute significantly to the success of a program (Stelter et al. 2021).

However, the weaknesses of the pre-experimental design discussed above (overestimation due to socially desirable answers and temporary effects) can and must be taken into account when interpreting the results.

## 6. Summary and Outlook

Despite the many measures by colleges and universities to increase the proportion of women in STEM courses, empirical evidence about their effectiveness is very patchy. Given the enormous challenge of applied evaluation research, this is not surprising. Ideal textbook methods cannot be applied 1:1 to the evaluation of STEM projects, since case numbers are usually very small and the long-term impact may not become apparent until years later. We have therefore developed an evaluation concept that best applies the scientific requirements for a valid evaluation and impact measurement of mentoring programs for female STEM students at colleges and universities. Our concept is based on the knowledge that career decisions are influenced by numerous factors. Some of these determinants or barriers to pursuing STEM studies or choosing a STEM career can be positively changed through targeted information and experiences from role models within mentoring programs. These interrelationships are presented in a scientifically based impact model. The evaluation concept thus consider that gender equality measures do not have a monocausal effect in principle, but rather contribute to the project goal via various paths. By observing the various partial effects on the level of outcome, it is also possible to make statements about the exact impact mechanism of a specific mentoring program. The program manager can use the evaluation results not only to promote the program, but also to adapt the content to the needs of the female students, and to develop the program as well as the mentees. Especially mediocre or very diverse feedback on the content and organization of the project (see indicators at the output level) give reason to critically reflect on the conception and the operational implementation.

The stronger the effect of the mentoring program on the diverse determinants of academic success and career choice, the greater the likelihood that mentees will actually decide to pursue a career in STEM fields as a result of participation. Changing all determinants in the desired way is not required to call a mentoring program successful. For one student,

getting a clearer idea of possible career paths may be enough, and, for another student, a better assessment of her self-efficacy will lead to the desired outcome.

In general, low case numbers and the individual effects by the mentors and trainers will make it difficult to derive general statements from one single evaluation of a specific mentoring program. Nevertheless, in order to generate more knowledge about the effects of programs to increase the proportion of women in STEM professions, there is no way around taking a closer and routine look at even small-scale projects and measuring their effects using standardized methods. The evaluation concept presented here is very flexible and can be used as a prototype for the evaluation of very different measures, also for younger female students. If necessary, additional indicators can be added at the outcome level. Our evaluation concepts hence contribute to gaining knowledge about the effects with STEM projects in the long term.

**Author Contributions:** Conceptualization, E.W.; methodology, E.W. and S.B.; formal analysis, E.W.; investigation, E.W. and S.B.; writing—original draft preparation, E.W. and S.B.; writing—review and editing, E.W.; visualization, E.W. and S.B.; supervision, E.W.; project administration, E.W.; funding acquisition, E.W. All authors have read and agreed to the published version of the manuscript.

**Funding:** This research was funded by the German Federal Ministry of Education and Research (BMBF) under grant number 01FP1715. The responsibility for all content supplied lies with the authors.

**Institutional Review Board Statement:** Not Applicable.

**Informed Consent Statement:** Not Applicable.

**Data Availability Statement:** Not Applicable.

**Acknowledgments:** We thank Andrea Löther (GESIS—Leibniz Institute for the Social Sciences, Cologne) for numerous and constructive discussions and thank the three anonymous reviewers for valuable comments on our manuscript.

**Conflicts of Interest:** The authors declare no conflict of interest.

## Notes

[1]   Of course, mentoring programs can not only impact participants, but also contribute to structural change within a university or faculty. Although these cultural changes are most valuable to women's opportunities in STEM, measuring these changes is far more difficult and goes beyond our goal.

[2]   The choice of the concrete questions is fundamentally important to gain valid results. Note, however, that the list of questions for the empirical measurement of self-efficacy and stereotypes should always be adapted to the concrete program, the specific target group of the mentoring program, and the cultural context. A universal valid questionnaire for all mentoring programs is not available, though. Herce-Palomares et al. (2022), Luo et al. (2021), and Verdugo-Castro et al. (2022) describe different methodologies to develop and validate instruments to quantify the concepts of self-efficacy and gender stereotypes in the context of STEM professions, and can serve as a blueprint for designing an adequate questionnaire.

[3]   The pre-tests within our pilot studies revealed that statements that formulate a very specific and certain intention better capture personal differences in terms of credibility of intention. We therefore followed the example of Fellenberg and Hannover (2006) and formulated very clear statements regarding the future studies and career plans.

[4]   This measure requires a matching of before and after responses on an individual level and reveals individual effects that might remain invisible in the group means.

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
