# Peer review of "Unlocking the Power of Mentoring: A Comprehensive Guide to Evaluating the Impact of STEM Mentorship Programs for Women"

_socsci, doi:10.3390/socsci12090508_

Round 1
Reviewer 1 Report
The article's topic, the impact evaluation of gender equality activities in higher education, is highly relevant. The paper seeks to improve the quality of evaluations through a theoretically founded and practical evaluation approach. The authors focus on mentoring programs for female STEM students, but the concept might apply to other activities.
The proposed evaluation concept is informed by evaluation theory, change theory and empirical studies on gender inequalities in STEM. The concept presents a theoretically informed solution to the problem that organizations that offer mentoring programs don't have the resources to evaluate the impact of the programs through long-term panel analysis. The evaluation concept focuses on several partial effects at different impact stages by applying change theory and a logical chart model. The selection of the partial effects is based on scientific findings on women in STEM fields. In the same way, the authors state that the impact – behaviour change – can be evaluated through the participants' intentions because studies prove intentions as a good predictor for behaviour.
In sum, the paper presents a theoretical and empirical founded evaluation concept which is at the same time, pragmatic and applicable for practitioners.
There are only minor points for revision:
While the evaluation design is well-founded, the concept of interpreting the empirical results is fuzzier (452-475). The reader would like to know more about criteria or the weighting of indicators to assess success. The authors remind to consider the evaluation design's weaknesses and remain cautious in interpreting the results. Nevertheless, there is the risk of relativizing bad results in some indicators when guidelines for an overall assessment are lacking.
The authors discuss well-informed the advantages and difficulties of different evaluation designs (370-425). They conclude that pre-experimental before and after surveys "provide quite valid information" and that "more valid evaluation designs are generally not applicable". This statement goes too far because planners of mentoring programs (or other gender equality activities) should at least check if they could apply experimental or quasi-experimental designs.
In the introduction, the authors state the majority of female engineering students are enrolled in UAS. Therefore, it would be surprising that mentoring programs' impact is not assessed regularly (43-45). The link between the statements is not logical.
Finally, I propose to explain more explicitly why the paper is relevant to an international journal. In the introduction (64-65), the authors refer to the lack in Germany without indicating the link to an international public.
Author Response
Dear reviewer,
I was very pleased to receive your constructive and well thought out comments. I am sure that the paper has gained in terms of comprehensibility and applicability. In the following I expound how I accommodated your comments. I hope that may revision of the paper addresses your concerns.
- "While the evaluation design is well-founded, the concept of interpreting the empirical results is fuzzier (452-475). The reader would like to know more about criteria or the weighting of indicators to assess success. The authors remind to consider the evaluation design's weaknesses and remain cautious in interpreting the results. Nevertheless, there is the risk of relativizing bad results in some indicators when guidelines for an overall assessment are lacking."
>> I understand that the reader or evaluator would like to have clear criteria for the success of a mentoring program. This does, however, not fit to the idea of the evaluation approach which tries to shed light on the various obstacle that female STEM-students have to deal with. I rather try to bring the reader or evaluator to the point of deciding whether the mentoring program has achieved the desired goals or not. In the revised version, I explicitly address the risk of relativizing bad results by pointing out that “undesirable changes should be taken as an opportunity to take a closer look at the suitability of the mentors (in terms of gender competence and the knowledge how to promote mentees) as well as the quality of the mentoring relationship, as these factors also contribute significantly to the success of a program (Stelter et al. 2021).” (see line 521-525). - "The authors discuss well-informed the advantages and difficulties of different evaluation designs (370-425). They conclude that pre-experimental before and after surveys "provide quite valid information" and that "more valid evaluation designs are generally not applicable". This statement goes too far because planners of mentoring programs (or other gender equality activities) should at least check if they could apply experimental or quasi-experimental designs."
>> I agree and I inserted two sentences emphasizing the benefits of additional information of non-participants. “Nevertheless, when designing a program or an evaluation, project managers and evaluators should explore all possibilities to collect complementary information on non-participants. Even if random assignment is not fully possible, these data provide valuable information on the selectivity of participants, which in turn might have implications for addressing the intended target group in the marketing.” (see line 467 – 473). - "In the introduction, the authors state the majority of female engineering students are enrolled in UAS. Therefore, it would be surprising that mentoring programs' impact is not assessed regularly (43-45). The link between the statements is not logical."
>> You are completely right. I skipped the two sentences and continue with the results of the meta-evaluation. - "Finally, I propose to explain more explicitly why the paper is relevant to an international journal. In the introduction (64-65), the authors refer to the lack in Germany without indicating the link to an international public."
>> I now mention two more international guides (National Center for Women & information Technology (2011) from the US and Kingsley (2020) from Australia). While all guides provide helpful insights on how to measure the effects of mentoring programs for female STEM students, our proposal stands out for its evidence-based foundation of impact relationships and evaluation methods. (see line 62 – 68).
Reviewer 2 Report
Passion is Helpful, Evaluation is Necessary. How to Assess the Impact
of Mentoring Programs for Female STEM Students
Comments
1. This paragraph presents an interesting topic on the evaluation of mentoring programs for female STEM students. The author notes that while mentoring programs are often carried out with passion, robust results on their effects are missing. The need for regular evaluations is emphasized, particularly to ensure an efficient allocation of resources toward gender balances in STEM. The proposed evaluation concept, which captures three levels of expected effects and derives a set of success indicators for each level, appears well thought-out and practical. The ability to observe the effect of a mentoring program directly after its end is a major advantage of the design.
2. Suggestion for a more compelling title:
a. Empowering Women in STEM: A Guide to Measuring the Success of Mentorship Programs
b. Empowering Women in STEM: The Importance of Assessing Mentoring Programs to Ignite Passion and Drive Success
c. Unlocking the Power of Mentoring: A Comprehensive Guide to Evaluating the Impact of STEM Mentorship Programs for Women
3. Some current studies on mentoring program for female STEM students are available. Considering updating your Literature Review.
4. You provide a well-reasoned explanation of the advantages of an immediate post-participation evaluation approach and how it can be used to identify the specific hurdles that mentoring programs can help female STEM students overcome. However, I suggest that you consider discussing limitations: while you mentioned some of the limitations of conducting a longitudinal evaluation of mentoring programs, it may be beneficial to discuss other potential limitations of the immediate post-participation evaluation approach and how they can be addressed.
5. One suggestion for improvement might be to provide more concrete examples of the specific obstacles that mentoring programs can help to overcome to make the concept more tangible for readers. Additionally, it would be helpful to provide more information on the types of data that would be collected and analyzed as part of the evaluation process to give readers a better understanding of how the evaluation concept works in practice.
6. The evaluation concept seems well thought-out and comprehensive, taking into account the challenges of evaluating STEM projects with small sample sizes and long-term impacts. The use of a scientifically-based impact model is a strong point, as it provides a framework for measuring the effectiveness of mentoring programs for female STEM students. The approach of considering the various partial effects on the level of outcome is also a sound strategy, as it allows for a better understanding of the impact mechanism of a specific program. The idea of using evaluation results to improve the program and adapt it to the mentees’ needs is also commendable. However, it would be helpful to have more information on the specific indicators and methods that will be used to measure the various partial effects and outcomes. Additionally, it would be beneficial to address the potential limitations and challenges of the evaluation concept and how they will be mitigated.
7. In addition to evaluating the short-term impact of the program, consider suggesting something about following up with the participants over the long term to determine the extent to which the program has contributed to their success in STEM fields.
Author Response
Dear reviewer,
I was very pleased to receive your constructive and well thought out comments. I am sure that the paper has gained in terms of comprehensibility and applicability. In the following I expound how I accommodated your comments. I hope that may revision of the paper addresses your concerns.
- Suggestion for a more compelling title:
>> thank you very much for the smart titles. - "Some current studies on mentoring program for female STEM students are available. Considering updating your Literature Review."
>> I carefully checked the current literature on meaningful evaluations of mentoring programs for female STEM students and I added the following studies (which are however mostly not brand-new): Beck et al. (2021), MacPhee et al. (2013), Reid et al. (2016), Stöger et al. (2016, 2017), Stout et al. (2011) and Strayhorn (2012). More recent publications have not come to my attention. If you have any concrete suggestions to me, I would be very grateful. Occasionally I also added more recent literature in other places (e.g. Hughes 2014, Miner et al. 2019, Rea 2015, Shaffer et al. 2013, Stelter et al. 2021, …) - "You provide a well-reasoned explanation of the advantages of an immediate post-participation evaluation approach and how it can be used to identify the specific hurdles that mentoring programs can help female STEM students overcome. However, I suggest that you consider discussing limitations: while you mentioned some of the limitations of conducting a longitudinal evaluation of mentoring programs, it may be beneficial to discuss other potential limitations of the immediate post-participation evaluation approach and how they can be addressed."
>> I agree that the post-participation evaluation is also not without any problems. However, I address possible aspects that may lead to bias in the data in a separate section: “Nonetheless, the survey results must be interpreted with caution. For example, the survey of female students, which often takes place directly after participation in the measure, increases the probability of socially desirable answers. This is especially true if the participants fill out the questionnaires in the presence of the project coordinators, whose project they may not want to rate badly. Even if the students do not intend it, the Hawthorne effect, which is caused by the special attention given to the participants, leads to an overestimation of the measured effects in a pre-post survey (Colbjørnsen, 2003). However, this bias is inherent in all surveys and can only be mitigated by de-signing a professional interview situation, ideally without interviewers who are somehow personally involved in the mentoring program. On the other hand, it is conceivable that the effect of the support measure is greater directly after the last event than at a later point in time, so that the long-term effects in particular may be overestimated.” (see line 453-464). - "One suggestion for improvement might be to provide more concrete examples of the specific obstacles that mentoring programs can help to overcome to make the concept more tangible for readers. Additionally, it would be helpful to provide more information on the types of data that would be collected and analyzed as part of the evaluation process to give readers a better understanding of how the evaluation concept works in practice."
>> To make the specific obstacles that women face more tangible, I added some sentences in section 2 (right after the noting that these challenges are well researched. (see line 123 – 134). Furthermore, I added a short section about the type of data that would be collected and how it could be analysed in section 5, right before I talk about the interpretation of the data. I therefore changed the title of the section into “Collecting and interpreting the data”. (see line 475-491). - "The evaluation concept seems well thought-out and comprehensive, taking into account the challenges of evaluating STEM projects with small sample sizes and long-term impacts. The use of a scientifically-based impact model is a strong point, as it provides a framework for measuring the effectiveness of mentoring programs for female STEM students. The approach of considering the various partial effects on the level of outcome is also a sound strategy, as it allows for a better understanding of the impact mechanism of a specific program. The idea of using evaluation results to improve the program and adapt it to the mentees’ needs is also commendable. However, it would be helpful to have more information on the specific indicators and methods that will be used to measure the various partial effects and outcomes. Additionally, it would be beneficial to address the potential limitations and challenges of the evaluation concept and how they will be mitigated."
>> I agree that the theoretical underpinning is more sophisticated than the guidance of empirical implementation. I therefore added some information about specific indicators (e.g. gender stereotype or self-efficacy, which often cannot be addressed by only one statement, such that multiple items are used in the questionnaire and then aggregated into an appropriate indicator). Furthermore, I describe three ways of comparing the before and after mentoring answers. (see line 475 – 491). The potential limitations of the evaluations concept and how they can be mitigated are addressed in section 4 (see line 453 – 464). We therefore decided not to repeat them one page later, but we explicitly refer to them in the final sentence of section 5. - "In addition to evaluating the short-term impact of the program, consider suggesting something about following up with the participants over the long term to determine the extent to which the program has contributed to their success in STEM fields."
>> In theory you are right. More data at different points in time always yield more insight. But, as far as I know the empirical literature on evaluation of mentoring programs, missing response to questionnaires is one of the biggest problems. That’s why we developed this evaluation concept which does not depend on long-term tracking of participants. Suggesting some sort of follow-up interviews would cast doubt on the validity of our concept. We therefore refrain from adding any comment on follow-up interviews.
Reviewer 3 Report
Authors presents an interesting work about evaluating STEM mentoring programs for women. However, it has several shortcomings.
Firstly, there are already several works in this line of research that are not included in the referenced bibliography. As an example, the authors could review these references:
Herce-Palomares MP, Botella-Mascarell C, de Ves E, López-Iñesta E, Forte A, Benavent X, Rueda S. On the Design and Validation of Assessing Tools for Measuring the Impact of Programs Promoting STEM Vocations. Front Psychol. 2022 Jun 27;13:937058. doi: 10.3389/fpsyg.2022.937058. PMID: 35859828; PMCID: PMC9291434.
Halim, Lama, Tuan Mastura Tuan Soh, and Nurazidawati Mohamad Arsad. "The effectiveness of STEM mentoring program in promoting interest towards STEM." Journal of Physics: Conference Series. Vol. 1088. No. 1. IOP Publishing, 2018.
National Academies of Sciences, Engineering, and Medicine; Policy and Global Affairs; Board on Higher Education and Workforce; Committee on Effective Mentoring in STEMM; Dahlberg ML, Byars-Winston A, editors. The Science of Effective Mentorship in STEMM. Washington (DC): National Academies Press (US); 2019 Oct 30. 6, Assessment and Evaluation: What Can Be Measured in Mentorship, and How? Available from: https://www.ncbi.nlm.nih.gov/books/NBK552763/
Habig, Bobby. "Practical rubrics for informal science education studies:(1) a STEM research design rubric for assessing study design and a (2) STEM impact rubric for measuring evidence of impact." Frontiers in Education. Vol. 5. Frontiers Media SA, 2020.
Secondly, the text does not clearly reflect the basis of the evaluation proposal. That is to say, there are various questions, some taken from other works and others new, without indicating how they have been designed. It is also not mentioned whether a validation process for the defined questions has been carried out and whether their validity has been measured regarding the problem being evaluated. The text also lacks information on whether a pilot test has been conducted to determine the effectiveness of the proposed approach.
I would recommend the authors to conduct a validation of the defined questions and a pilot test, using the recommended references as an example. Additionally, it would be beneficial to have a table or graph that clearly correlates the problems to be evaluated with the corresponding measuring question.
I hope authors would clarify my concerns and solve the remarked issues to achieve a high-quality article appropriate to journal level.
Author Response
Thank you very much for your valuable comments. We are sure that the paper has gained in terms of comprehensibility and applicability. In the following we expound how we accommodated your comments. We hope that our revision of the paper addresses your concerns.
1) Additional literature
Halim et al. (2018) analyse the impact of a mentoring program in Malaysia on the interest in STEM subjects. They use a post-test control group design and focus on differences in interest. In my view the evaluation design of this study is not convincing for two reasons. First, the post-test control group design suffers from strong selection effects in case of voluntary programs (in this case, only 25% of the participants were randomly assigned). Second, focussing on change in interest is not enough to assess the overall change in the propensity to decide for a STEM career (which is definitely the more relevant indicator). There are, however, many other relevant determinants of this indicator – especially for women – that should be considered for a comprehensive evaluation of a mentoring program (as I expound in detail in my paper). I therefore refrain to refer to this study.
The publication of the National Academies of Sciences, Engineering, and Medicine (2019) was already referred to in the introduction (see p. 2, line 70/71).
Habig (2020) defined rubrics to classify the research design and the impact of informal science education programs (ISE) and uses the knowledge of experts to assess the validity of studies analysing these programs. Basically, he does a kind of meta-evaluation in the field of ISE programs. Even if the motivation of his study (huge variety of studies with missing information about sufficient evidence) is very similar to my motivation to develop a sound and easy to apply evaluation design for mentoring programs, I do not think that this publication significantly contributes to my paper. First of all, Habig (2020) provides very general rubrics which can be used to design an evaluation. In contrast, our paper proposes a very concrete evaluation concept. Second, Habig (2020) focusses on ISE programs and we work on mentoring programs for female students. [The added value of mentoring programs lies in the establishment of a personal relationship with a female role model as well as personality development opportunities (for breaking down gender stereotypical thinking patterns and strengthening personal self-efficacy of female students). ISE programs, on the other hand, focus on strengthening interests and awareness in STEM disciplines – regardless of the sex (National Research Council, 2009; Young et al., 2017; Habig et al., 2018). The intersection of these two approaches is very small, though.]. Therefore, we do not refer to this reference in our paper.
Herce-Palomares, Maria Pilar, Carmen Botella Mascarell, Esther de Ves, Emilia López-Inesta, Anabel Forte, Xaro Benavent, and Silvia Rueda. 2022. On the Design and Validation of Assessing Tools for Measuring the Ompact of Programs Promoting STEM Vocations, Frontiers in Psychology 13:937058. doi.org/10.3389/fpsyg.2022.937058
--> This paper is integrated in footnote 2. See our explanations in the context of validation.
2) Validation and pilot study
Of course, the choice of the concrete questions is fundamentally important to gain valid results. This is especially true for questions aiming to measure self-efficacy and gender stereotypes (see 2.3 and 2.4 in figure 1). From our point of view, however, a universal guideline with concrete questions for the empirical measurement of self-efficacy and stereotypes makes no sense here, because the questions should always be adapted to the concrete program and the specific target group of the mentoring program. As Luo et al. (2021) expound, there exists very specific approaches to measure self-efficacy in different disciplines. Depending on the mentoring program under consideration, the evaluator should rather check-out discipline-specific and target group specific measurement concepts instead of one universal concept that I validated in the context of one specific mentoring program. Also Verdugo-Castro et al. (2022) point to the fact that very specific instruments are required to properly determine students’ stereotypes toward STEM subject. Replicating existing questionnaires is often not possible, either because the complete list of items is not available or the instruments were contextualised in other research questions, other target groups or even other cultures. Consequently, they develop their own methodology to design and validate the instrument to measure gender stereotypes in their specific context. We therefore conclude that recommending a universal set of questions to assess self-efficacy and gender stereotypes of participants in any mentoring program does not do justice to the complexity of the matter.
To properly appreciate the challenges in measuring these two concepts, we now address the need for validation and possible approaches, as recommended by you, in section 3.2, where we discuss the effects of self-efficacy and gender stereotypes. We decided to put this information into the footnotes (see footnote 2), because this issue does not directly contribute to our main contribution, which is the theoretically founded evaluation concept. It is, however, a valuable information for the application of the concept.
Measuring intentions (see 3.1 in figure 1) is less complicated, but may be not self-evident either. We therefore added a footnote (see footnote 3) summarizing our experiences from the pre-test in our pilot studies. Furthermore, we point out in section 2, that every questionnaire must be adapted to the specific context of the mentoring program and should apply validated instruments to measure the relevant indicators. Overall, the requirement for validation is thus explicitly taken into account.
Questions to measure the support and guidance during studies as well as the knowledge about job profiles (see 2.1 and 2.2 in figure 1) are much easier to develop. The questions we propose in the paper are either tested by others (e.g. Höppel 2016) or by ourselves. We now expound in the introduction that the development of the evaluation concept was accompanied by three empirical pilot studies in which the concept was applied to very different mentoring programs at universities of applied sciences and continuously improved (see p. 2, line 95ff). Within these pilot studies, we did pre-tests of the corresponding questionnaire with up to 15 students in different fields and ages.
We hope that we addressed your suggestions in an appropriate manner.
Round 2
Reviewer 3 Report
The authors have only partially taken into account the proposed recommendations in order to clarify the proposed evaluation process.
It is essential that the authors clarify this aspect and a footnote is not useful.
A quality article always needs a section on methodology, or materials and methods. It is essential to add this section and clearly describe the methodology and process used in the design and validation of the presented instrument.
Author Response
Dear reviewer,
thank you very much for the quick response to our second revision. Unfortunately, you did not fully accept the way we accommodated your recommendations.
We argue that a universal guideline with concrete questions for the empirical measurement of stereotypes and attitudes makes no sense here, because questions should always be adapted to the concrete program and the specific target group of the mentoring program. We refer to high quality validation studies from the literature.
In your short answer, you do not address our substantiated concerns regarding the development and validation of a universal set of questions. Your criticism that the paper needs a chapter on methodologies is not convincing from our point of view, since the development of the evaluation concept is clearly derived from the common methods of evaluation research. Due to the sweeping nature of your criticism, is seems almost impossible to us to implement your wishes satisfactorily.
To accommodate your concerns, we could also eliminate all the questions we offer to measure the different success factors. The core of our contribution - the development of a theoretically sound and comprehensive evaluation approach - would be almost unaffected. From our point of view, it would nevertheless be a loss, since the application-oriented reader would no more receive assistance for the empirical implementation of the evaluation concept through our examples.
Round 3
Reviewer 3 Report
The authors have not provided any changes to meet the request that it is essential, in a scientific article, to adequately describe the methodology used.